# Transcriptomic Differentiation of Phenotypes in Chronic Rhinosinusitis and Its Implications for Understanding the Underlying Mechanisms

**DOI:** 10.3390/ijms24065541

**Published:** 2023-03-14

**Authors:** Jure Urbančič, Tanja Košak Soklič, Ajda Demšar Luzar, Irena Hočevar Boltežar, Peter Korošec, Matija Rijavec

**Affiliations:** 1Department of Otorhinolaryngology and Cervicofacial Surgery, University Medical Centre Ljubljana, Zaloska 2, SI-1000 Ljubljana, Slovenia; 2Faculty of Medicine, University of Ljubljana, Vrazov trg 2, SI-1000 Ljubljana, Slovenia; 3Laboratory for Clinical Immunology and Molecular Genetics, University Clinic of Respiratory and Allergic Diseases Golnik, Golnik 36, SI-4204 Golnik, Slovenia; 4Faculty of Pharmacy, University of Ljubljana, Aškerčeva 7, SI-1000 Ljubljana, Slovenia; 5Biotechnical Faculty, University of Ljubljana, Jamnikarjeva 101, SI-1000 Ljubljana, Slovenia

**Keywords:** sinusitis, sequence analysis, transcriptome, RNA, phenotype

## Abstract

Chronic rhinosinusitis (CRS) is a multifaceted disease with variable clinical courses and outcomes. We aimed to determine CRS-associated nasal-tissue transcriptome in clinically well-characterized and phenotyped individuals, to gain a novel insight into the biological pathways of the disease. RNA-sequencing of tissue samples of patients with CRS with polyps (CRSwNP), without polyps (CRSsNP), and controls were performed. Characterization of differently expressed genes (DEGs) and functional and pathway analysis was undertaken. We identified 782 common CRS-associated nasal-tissue DEGs, while 375 and 328 DEGs were CRSwNP- and CRSsNP-specific, respectively. Common key DEGs were found to be involved in dendritic cell maturation, the neuroinflammation pathway, and the inhibition of the matrix metalloproteinases. Distinct CRSwNP-specific DEGs were involved in NF-kβ canonical pathways, Toll-like receptor signaling, HIF1α regulation, and the Th2 pathway. CRSsNP involved the NFAT pathway and changes in the calcium pathway. Our findings offer new insights into the common and distinct molecular mechanisms underlying CRSwNP and CRSsNP, providing further understanding of the complex pathophysiology of the CRS, with future research directions for novel treatment strategies.

## 1. Introduction

Chronic rhinosinusitis (CRS) is a debilitating chronic inflammatory disease affecting around 5–12% of the population [1]. Dividing CRS into two significant phenotypes, CRS with nasal polyps (CRSwNP) and CRS without nasal polyps (CRSsNP), does not account for all the variability in the clinical course, global endotype presentation, responsiveness to therapy or treatment outcomes [1,2,3,4]. CRSsNP manifests as type 1 (T1) or type 3 (T3) inflammation, predominantly without eosinophilia. Type 2 (T2) inflammation is found primarily in CRSwNP with eosinophilia [5,6,7]. Furthermore, overlapping cytokine profiles may be found even in the same geographical area [4,8,9]. Due to the beforementioned variability, CRS is considered a range of distinct conditions with a similar clinical presentation but different underlying inflammatory mechanisms [1,7].

Dysregulation of the immune response is essential to our present understanding of CRS. Identifying those mechanisms and their understanding might bring better insight into the multifaced illness. Using transcriptomic signatures to identify exact disease processes could shed light on complex molecular interactions in CRS [10,11]. The most studied/plausible/identified candidates are the genes involved in inflammation and proinflammatory response, genes involved in remodeling the sinonasal tissue, genes involved in disrupting the epithelial barrier, and other genes with a less clear role in CRS [2,6,12]. Researchers using various techniques have reported specific genes involved in CRSwNP and CRSsNP [12,13,14]. The available data still seems insufficient, and additional research is needed to complete and confirm the role of specific genes, identify biomarkers of CRS or even find causal relationships [3,15]. Common pathways of T2 inflammation in asthma or atopic dermatitis may help identify cardinal processes [16,17]. Rigorous study design in a clear-cut phenotype is a prerequisite to successfully interpreting big data in any genetic study [2,18]. This study aimed to shed new light on the biological pathways and markers of different phenotypes in CRS (CRSsNP and CRSwNP) compared to healthy individuals, with a transcriptomic analysis of genes, and to share the publicly available dataset to fulfill the gaps in understanding of the pathophysiology of CRS. It is, to our knowledge, one of a more phenotype-oriented dataset. The study aims to illuminate the underlying molecular mechanisms in CRSwNP and CRSsNP by using the differences in gene expression compared to the controls. Herein, we have undertaken a comprehensive transcriptomic assessment to identify key genes and pathways underlying different CRS phenotypes in central and eastern European populations.

## 2. Results

### 2.1. Characteristics of the Study Cohort of Patients with CRS and Controls

The whole cohort involved 488 patients with CRS, and 37 controls. A total of 123 patients were eligible to be enrolled in the study. Due to exclusion criteria, 87 tissue samples were taken. Gene analysis was carried out on 22 subjects with CRSwNP, 11 with CRSsNP and 15 control subjects (Table 1). Their age and sex did not differ significantly. Patients with CRS (vs. controls) more often reported a family history of CRS and had a significantly higher incidence of allergies (*p* = 0.03), asthma (*p* = 0.02), and gastroesophageal reflux disease (GERD) (*p* = 0.03). The CRSwNP group also showed predominant local eosinophilia in the tissue samples, with a mean number of 30 eosinophils in high-power field (HPF). We found no difference in the number of previous surgeries, time interval from the last surgical intervention, or Sinonasal outcome test 22 (SNOT-22) QoL score between subjects with CRSwNP and subjects with CRSsNP. The clustering using tissue eosinophilia levels into non-T2 and T2 subtypes significantly differed between phenotypes [1,7]. CRSwNP had a T2 endotype (100%) exclusively, and CRSsNP was predominantly a non-T2 subtype (91%) (Table 1) [1,19,20].

### 2.2. Common and Phenotype-Specific Transcriptional Alteration in CRS

We first compared differentially expressed genes (DEGs) in tissue samples of patients with CRSwNP/CRSsNP and control individuals, and found a high correlation between CRSwNP and CRSsNP (r = 0.637, *p* < 0.001; Figure 1). However, differences were also observed, as 375 CRSwNP-specific DEGs and 328 CRSsNP-specific DEGs were identified (Figure 1). Several unique phenotype-associated DEGs with high differential expression values are presented in Table 2. Furthermore, we also identified 75 DEGs CRS-and phenotype-specific genes (Figure 1; details in Appendix A). To ensure the phenotype-endotype validity and confirm the clinical endotyping into T2 and non-T2 inflammation, the expression of cytokine-related genes associated with type 1 (T1), type 2 (T2), or type 3 (T3) inflammation [2] is presented in Table 3.

### 2.3. Transcriptomic Alterations in CRS Are Associated with Immune Response Modulation, Cellular Movement, Bone Homeostasis, and Inflammation

DEGs represent only a part of the transcriptomic alteration analysis. To illustrate interactions, biological context, and bio-function among the top DEGs of different endotypes in CRS, a network and pathway analysis was performed using QIAGEN’s Ingenuity^®^ Pathway Analysis (IPA^®^, QIAGEN, www.qiagen.com/ingenuity (accessed on 20 November 2021)). Enrichment of DEGs shows immune regulation, cellular movement, and inflammation, suggesting their regulation is the most critical pathway in CRS. However, differences are notable between phenotypes, as in CRSwNP immune regulation through NF-kβ, Toll-like receptor signaling, and crosstalk, HIF1α regulation, Th2 pathway, IL-6, and IL-15 pathways recruitment, and the regulation of various immune cells. On the other hand, in CRSsNP, NFAT regulation of immune response, calcium-induced apoptosis of T cells were identified as highly activated pathways. CRSsNP specific were SPINK1 pancreatic cancer pathway and LXR/RXR activation (Figure 2; Details in Appendix A). 

An analysis using QIAGEN’s Ingenuity^®^ Pathway Analysis (IPA^®^, QIAGEN, www.qiagen.com/ingenuity (accessed on 20 November 2021)) uncovered common upstream regulators triggering the inflammatory processes and regulating the immune pathways, known innate immunity response molecules that have already proven to be involved in CRS, bacterial product (Staphylococcus aureus enterotoxin), and T-cell regulators. CRSwNP-dominant regulators are engaged in T2 inflammation or eosinophilia, such as ERBB, TNF pathway, inflammatory and host defense, cytokine regulators and their cytokines, such as CSF2 or IL4, and angiogenetic factors. CRSsNP expresses molecules involved in NFAT regulation, T-cell regulation, and stress-response genes (Figure 3, details in Appendix A).

CRS-associated bio functions identified in the IPA analysis include cell-to-cell signaling, cellular movement, cellular function, the maintenance of inflammatory response, and immune-cell trafficking. Downregulated CRS-associated DEGs were enriched in biological functions related to the infection of mammalia, immunodeficiency, and parasitic infection. CRSwNP-specific DEGs were enriched in processes related to eosinophilia and the quantity of neutrophils. At the same time, CRSsNP-specific DEGs were involved in functions related to inflammation and apoptosis (Figure 4, details in Appendix A).

## 3. Discussion

This study presents a comprehensive assessment of CRS-associated nasal tissue transcriptome of a cohort of well-characterized individuals. Common key genes and pathways involved in CRS include dendritic cell maturation, the neuroinflammation pathway, and the inhibition of matrix metalloproteinases (Figure 2, Appendix A). Furthermore, specific elements have also been identified. CRSwNP involved NF-kβ pathways, Toll-like receptor signaling, HIF1α regulation, and the Th2 pathway, whereas CRSsNP involved the NFAT pathway and changes in the calcium pathway. These findings offer new insights into the molecular mechanisms underlying CRSwNP and CRSsNP. They enhance our understanding of CRS and may help aim future research.

Due to heterogeneous phenotypes and various endotype dominance, there is still a lack of basic knowledge regarding the differences in the airway-tissue cell transcriptome of healthy individuals and patients with CRS [3]. The patient’s or the disease’s clinical characteristics or cytokine expression offer limited insight into the essential paradigm of T2 and non-T2 inflammation, but cannot reliably point out their pathophysiological underpinnings [2,5,22]. 

Identifying differentially expressed genes in different phenotypes of CRS and healthy subjects may suggest novel biological insight into the molecular mechanisms involved [23]. CRS-associated genes are numerous, and can be divided into the immune-system-related, barrier- and structural-related, bacterial-carrier-related, or others [3]. Some genes are supported by similar results from more than one study, while others may still lack sufficient evidence. CRS-associated gene-pool meta analysis by Fokkens et al. and Orlandi et al. defined raw-data similarity using a head-to-head comparison of genes and references [1,3]. 

Specific DEGs emphasize similarities and differences in the pathophysiological processes of both phenotypes. Commonly upregulated genes include *secretoglobin 1 C1 SCGB1C1,* confirming previously published data [24]. A difference in fold change may also imply a different role in the pathophysiology of both phenotypes. Nevertheless, they may both lack downregulation through IFN-γ (Appendix A). The *SCGB1C1* may be regarded as part of the host response protection against the common cold, as reviewed by Orysiak et al. [25].

Interestingly, this was downregulated in athletes with a below-mean IgA and deletion involved in susceptibility to psoriasis [25,26]. The late *cornified envelope 3D gene LCE3D* was distinctly downregulated in CRSsNP. It is one of the proposed targets for the therapy of atopic dermatitis (AD), as a key upregulated gene [27].

Our study has shown upregulated bone-homeostasis genes—*human bone sialoprotein gene IBSP* (generally upregulated), *extracellular phospoglycoprotein MEPE*, and *interferon-induced transmembrane protein 5 IFITM5* (both upregulated only in CRSsNP), which may play a potential role in viral infection [28,29]. We have also confirmed the upregulation of the *lipopolysaccharide-binding protein LBP* in CRSsNP. *LBP* is a part of the cytokine induction pathway in immune cells, and is involved in the acute-phase immune response to gram-negative bacteria [30]. It interacts with the CD14, also confirmed in our data as an activated upstream regulator in both phenotypes (Appendix A). CD14 is an essential receptor for the innate immune system, and was previously described as upregulated in CRS by Yao et al. [31]. Our study established *C-C motif chemokine ligand 7 CCL7* as dominantly overexpressed in CRSwNP. It is a part of the chemokine-stimulating pathway associated with macrophages [7,32]. Brunner et al. report it as a driver of TNF-dependent inflammation in psoriatic skin [33]. TNF is also one of the upstream regulators our data has shown to be activated in both CRS phenotypes, but it predominates in CRSwNP (Appendix A). The intricate interaction of the proinflammatory cytokine family might be different in both phenotypes, and coherent with some previously published data [34,35]. The most-upregulated unique gene for CRSwNP in our study was the *calcium-activated chloride channel regulator 1 CLCA1*, connected to the mucus cell metaplasia of chronic inflammatory airway diseases [36]. It is also highly upregulated in mouse and human models that respond to IL-13 and STAT6. They appear to be pivotal in mucus overproduction [37,38,39]. We have confirmed the high expression of the *interleukin 13 IL-13* gene and IL-13 as a highly active upstream regulator in CRSwNP (Appendix A), but the evidence of differences in the role of STAT6 as an upstream regulator in both phenotypes is unclear. Our data suggest only moderately higher activation of STAT6 in CRSwNP (Appendix A). Furthermore, we have seen significant activation of the STING1 upstream regulator in CRSwNP. Thus, the previously proposed model of STING-downregulation and STAT6-upregulation signaling may fit only in some endotypes of T2 CRS [40]. Our data confirms higher activation of IFN-β upstream regulator in CRSwNP and no inhibition of IFN-β in all subtypes of CRS, as stated by Hwang et al. [41]. The expression of the *interferon-gamma IFN-γ* gene is higher in CRSsNP, confirming the evidence of Stevens et al., showing IFN-γ as a marker of T1 inflammation [6]. (Appendix A). Similarly to Karatzanis et al., our data may confirm the vascular endothelial growth factor VEGF as an activated T2 upstream regulator in the CRSwNP phenotype [42]. However, subunit A (VEGFA) has been activated as an upstream regulator in CRSsNP and CRSwNP. Therefore, part of our data suggests similarities with cluster 3 of Divekar et al., showing a mixed CRSwNP/CRSsNP group with VEGF-dominant biomarkers (Appendix A) [43].

Our study unraveled the upregulated *adhesion G protein-coupled receptor G7 ADGRG7,* with an unknown role in CRS. It is also unique to CRSwNP, and probably involved in cellular adhesion, an integral component of the plasma membrane [44]. *B lymphocyte chemoattractant CXCL13* is upregulated only in CRSsNP, revealing a similar pattern as that indicated by Klingler et al. for the T3 endotype [7]. Our results show upregulated *opiorphin prepropeptide OPRPN* and *pannexin 3 PANX3* genes involved in bone formation, but only in CRSsNP. In previous studies, *OPRPN* was associated with better survival in oral squamous carcinoma and as a pain-relieving agent in the eye [45,46]. PANX3, on the other hand, represents a channel in cell-to-cell communication and migration that affects the cytoskeleton [47]. Additionally, PANX3 has a role in the recruitment of CD4+ T cells, neutrophils, and macrophages in mice [48]. 

Downregulated CRS genes were physical-barrier protein genes of the *calcium-binding protein* A7 *S100A7* family and the *matrix metallopeptidase 9 MMP9*, extensively related to barrier defects in both phenotypes of the CRS [49,50]. *Transmembrane proteinase TMPRSS11B* has been detected in various tissues, including the airways, but its function remains undetermined [51]. We have also seen the downregulation of additional NF-kβ-linked family representatives from the IL-1 superfamily, *IL36A*, *IL36γ,* and *IL36RN* (receptor antagonist) genes. Downregulated IL36A can activate NF-kappa-B and MAPK signaling pathways [52]. IL1RL2 mediates IL36γ’s activity, but a different mechanism may be involved with an expression, similar to the controls. The IL36/IL1 family may also have a role in tissue remodeling [53]. IL36RN (receptor antagonist) inhibits the activation of NF-kβ by interleukin IL1, and has been implicated as a protective element in cutaneous infections [52]. Our data show substantially higher downregulation of *IL36RN* gene in CRSsNP. *Interleukin 1β IL1B* was an overexpressed gene and activated upstream regulator in both phenotypes. Its role in CRSsNP seems elusive, and might be explained as an intrusion of the T2 endotype in the CRSsNP subgroup [7,54]. In contrast, the IL1A upstream regulator was activated solely in CRSwNP, and is firmly connected to eosinophilic CRSwNP (Appendix A) [55]. Karjalainen et al. have explained the correlation between IL1A and asthma in CRSwNP [56]. 

We have also confirmed a series of downregulated antimicrobial genes such as *lacritin LACRT,* the *proline-rich protein Haell subfamily 2 PRH2* [57,58], or immune-response-inducing *lipocalin 1 LCN1* [59]. The latter was also confirmed in eosinophilic CRS by Brar et al. [18]. CRSwNP patients have a downregulated *secretagogin SCGN* gene, proven to be involved in early-onset inflammatory bowel disease; the alteration of their gene expression might reveal modified innate immunity, thus also showing a possible genetic basis for the inflammatory reaction [60]. 

Our data confirms downregulated *prolactin-induced protein PIP* in CRSwNP. It has a role in binding to CD4 receptors on lymphocytes and Staphylococcus sp. bacteria [61,62]. Both have been implicated in the pathophysiology of CRS and other airway diseases [63]. *Cornulin gene CRNN* was the most downregulated gene in CRSsNP. It was previously found to be upregulated in atopic dermatitis and eosinophilia [64]. It could be consistent with the lack of T2 inflammation pathways in CRSsNP, similar to *LCE3D*. Specific downregulated barrier genes such as *desmoglein-1 DSG1* have been previously implicated in CRSsNP pathophysiology [65]. Other downregulated genes in CRSsNP were the *apoptosis-related protein gene CRCT1* [66], *barrier protein IVL* [67], and *antiprotease gene SPINK7*, being key inhibitory regulators of the inflammatory response, barrier function, and differentiation, respectively [68]. To an extent, our data correlate with Huang et al.’s findings regarding ferroptosis genes in CRS, by showing a pattern of upregulation of genes for *nitric oxide synthase 2 NOS2*, *arachidonate 5-lipoxygenase ALOX5* and the upstream regulator TFRC [69]. Our study implies the role of the upstream regulator SNAI1 in epithelial-to-mesenchymal transition (EMT) via influence on the process of ferroptosis (Appendix A) [70]. EMT is a crucial process in CRS, as has been proposed by various other authors [1,54,71]. 

The limitation of our study is still a relatively small sample size, due to rigid inclusion criteria and the adequacy of the RNA extracted. Similarly to other transcriptome-based studies, we have not directly addressed possible discordances in transcriptomics and proteomics [55]. Another potential limitation is the biopsy subsite heterogeneity in gene expression [10,72]. Our study was based upon a robust uniform-treatment regime, comparable data for phenotypisation, and clinical endotypisation as a prerequisite to comprehensively assess the transcriptome of nasal tissues in CRS (CRSsNP and CRSwNP). The validity of the groups was further established with cytokine-related DEGs (Table 3). We strived to achieve high-quality tissue from disease-wise appropriate nasal subsites in CRS patients and controls. Therefore, we carried out a protocol to reliably identify critical genes and pathways common to, and distinct from, CRSwNP and CRSsNP. 

## 4. Materials and Methods

### 4.1. Study Subjects

The study was designed as a prospective cohort study of 488 patients with CRS diagnosed and treated at a tertiary institution in Slovenia, responsible for managing the most burdensome patients. Ninety-eight eligible patients with a clear-cut phenotype of primary CRS (CRSwNP and CRSsNP) were identified [1,2]. All required surgical treatment and were older than 18 years. Patients with significant comorbidities such as aspirin-exacerbated respiratory disease (AERD), active smoking (smoking in the last year), a recent history of treatment with oral steroids or antibiotics (preoperative), confounding benign or malignant tumors, and a history of autoimmune diseases were excluded (n = 390). All patients had signed a written consent after a detailed explanation of the study design, the handling of their tissue samples, and the expected results, including the lack of direct and immediate effects of participation and study results on their treatment. The study was conducted according to the Declaration of Helsinki, and the National Ethics Committee approved the study (0120-297/2017/3 from 20 November 2017). 

The control group’s inclusion criteria were a requirement for nasal surgery other than CRS or allergic rhinitis (AR) and an age of more than 18 years. All had previously received maximal medical therapy consisting of intranasal fluticasone propionate drops (CRSwNP) or mometasonfuroat nasal spray (CRSsNP). Prolonged antibiotic treatment with doxycycline or macrolides or amoxicillin with clavulanic acid was also implemented during the preoperative workout, but not for at least a month before the tissue biopsy. Systemic corticosteroids were also not administered at least six months before the biopsy [1,19].

Exclusion criteria for controls were comorbid asthma, AR, benign or malignant tumor of the nose or paranasal sinuses, history of autoimmune diseases, and smoking. A detailed flowchart of patients and controls with inclusion parameters is presented in Figure 5. Sampling was performed on 34 patients with CRSwNP, 28 with CRSsNP, and 25 controls, during scheduled routine surgery from December 2017 to December 2019. Several rhinosurgeons carried out the collection of samples. The type of surgery and amount of sinonasal resection was chosen according to disease specificity. Patients were treated the same as patients not enrolled in the study. Clinical data were collected from an institutional database. Patient details including a family history of CRS, allergies (positive prick tests or positive total IgE), asthma diagnosed by a pulmonologist, history of gastrointestinal reflux disease or its therapy, subjective assessment of smell, and objective evaluation using Sniffin’Sticks 12 test (Burghart GmbH, Wedel, Germany) are presented in Table 1. Local eosinophilia was assessed with a cut-off count of 10 per HPF and peripheral eosinophilia cut-off of more than 250 eosinophils/µl in peripheral blood [1]. The number of past surgeries and the time from the last surgery were calculated. The need for two or more courses of systemic steroids in the past year was assessed. As well as computer tomography (CT) findings in the nose and paranasal sinuses using the Lund–Mackay scoring system [73], endoscopic evaluation using a modified Kennedy–Lund score [21,74], total polyp score 0–4, VAS score of CRS-related problems (0-no complaints, 10-worst complaints), and a validated Slovenian SNOT-22 score, were evaluated [75]. To ensure the validity of patient selection, a clinical clustering in T2 and non-T2 subtypes was carried out using the available clinical characteristics of the T2 endotype, primarily local tissue and blood eosinophilia [1,6,7,19].

### 4.2. Sampling Site and Biopsy

Exact locations were chosen, considering intranasal tissue differentiation [10,72]. In patients with CRS, the sampling site was the sinonasal mucosa of the middle meatus or adjacent sinonasal region. In CRSwNP, the accurate sampling site was the distal part of the polyp, the base of the polyp in the mucosa of the anterior ethmoids, and in CRSsNP, the diseased mucosa of the middle meatus, the anterior ethmoid or bulla ethmoidalis. In healthy controls, samples were taken from the same middle meatus area, the medial side of the uncinate process’s horizontal or vertical part. All samples were taken under direct endoscopic visualization, using 0-degree or 30-degree optics (Karl Storz, Tuttlingen, Germany, or Olympus, Hamburg, Germany). The collected sample was divided into two subsamples and separately stored in RNAlater solution (Qiagen, Hilden, Germany) at −40 °C until RNA isolation. At the same time, an additional tissue sample was collected from the same site for histopathological examination or staging of the CRS. This step was omitted in healthy controls without CRS.

### 4.3. RNA Sequencing

#### 4.3.1. Processing of Tissue Samples and RNA Isolation

Total RNA was extracted using the miRNeasy Mini Kit (Qiagen, Hilden, Germany) and TissueLyser (Qiagen, Hilden, Germany), according to the manufacturer’s instructions. The purity of the isolated RNA was assessed using the NanoDrop 2000c spectrophotometer (Thermo Fisher Scientific, Waltham, MA, USA), and integrity was determined on the Agilent 2100 Bioanalyzer, using the RNA 6000 Nano kit (Agilent Technologies, Santa Clara, CA, USA). All samples used for library preparation had RNA integrity values greater than 6.0. Extracted RNA samples were stored at −80 °C before further processing. An entire graphical-study flowchart is presented in Figure 6.

#### 4.3.2. Sample Library Preparation and Sequencing

RNA sequencing (RNAseq) was performed on the Illumina NovaSeq6000 platform with a per-sample target of 15 million 100 bp paired-end reads, by Macrogen Inc. (Seoul, Republic of Korea). According to the manufacturer’s protocol, sequencing libraries were prepared using the TruSeq Stranded Total RNA Library Prep workflow with Ribo-Zero Globin (Illumina, San Diego, CA, USA).

#### 4.3.3. RNAseq Data Analyses

RNA-Seq data were analyzed using the CLC Genomics Workbench (v20.0.4, Qiagen Bioinformatics) as described previously [76]. Briefly, quality trimming excluding low-quality reads, adapter trimming, alignment with *Homo Sapiens* (GRCh37.p13), and global TMM normalization of the reads, were performed. The CLC Genomics tool, applying multi-factorial statistics based on a negative binomial GLM for differential expression in two groups and the Wald test against the control group, were used to assess differences in gene expression. Gene-expression values were reported as RPKM (reads per kilobase of transcript per million mapped reads) and TPM (transcripts per million) [76]. Genes with a false discovery rate (FDR) of 0.05 or less and an absolute log2 fold change value of 1 or more were deemed differentially expressed.

#### 4.3.4. Functional and Pathway Analysis

The functional analyses, which included diseases and functions, canonical pathways, and upstream transcriptional regulators, were generated using QIAGEN’s Ingenuity^®^ Pathway Analysis (IPA^®^, QIAGEN Redwood City, www.qiagen.com/ingenuity (accessed on 20 November 2021)).

#### 4.3.5. Statistics

Statistical analysis and data preparation was performed using SPSS statistical software version 20 (SPSS Inc., Chicago, IL, USA), GraphPad Prism 9 (GraphPad Software, Boston, MA, USA), and Microsoft Excel (Microsoft, Redmond, WA, USA). The Kruskal–Wallis test, Pearson chi-square Monte Carlo, Fisher exact test, and Mann–Whitney U test were used. A *p* value of less than 0.05 was considered significant.

## 5. Conclusions

This study identifies common and distinct genes and pathways involved in CRS. For the first time, we have identified new CRS-related genes such as ADGRG7, probably reflecting cellular-adhesion elements, LACRT, IBSP, MEPE, and IFITM5, probable antimicrobial genes with a potential role in viral infections. CRNN may be a link to other biologically related diseases such as atopic dermatitis or eosinophilia. The last group included genes with unknown functions, such as TMPRSS11B or PRR4. Our findings offer new insights into the molecular mechanisms underlying CRSwNP, with the involvement of the NF-kβ pathway, HIF1α regulation, and Th2 pathway, or the NFAT pathway and changes in the calcium-induced apoptosis in CRSsNP. Our findings augment our understanding of CRS and suggest implicit future research directions to elucidate candidates for novel treatment strategies.

## Figures and Tables

**Figure 1 ijms-24-05541-f001:**
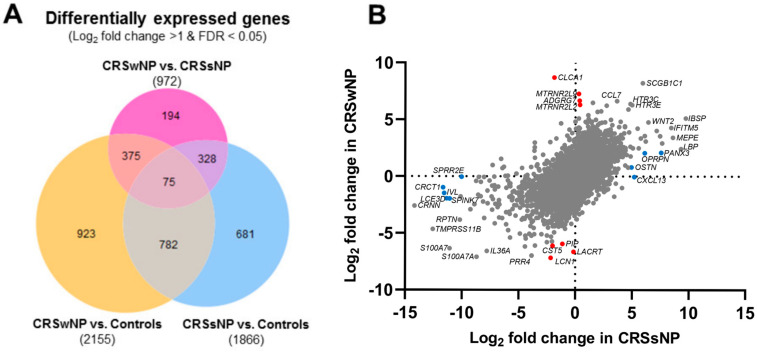
Venn diagram (**A**) and differential gene expression (**B**) in CRSwNP, CRSsNP, and controls. The proposed unique phenotype-associated DEGs with the highest differential expression in one of the phenotypes are presented in red (●) for CRSwNP and blue (●) for CRSsNP.

**Figure 2 ijms-24-05541-f002:**
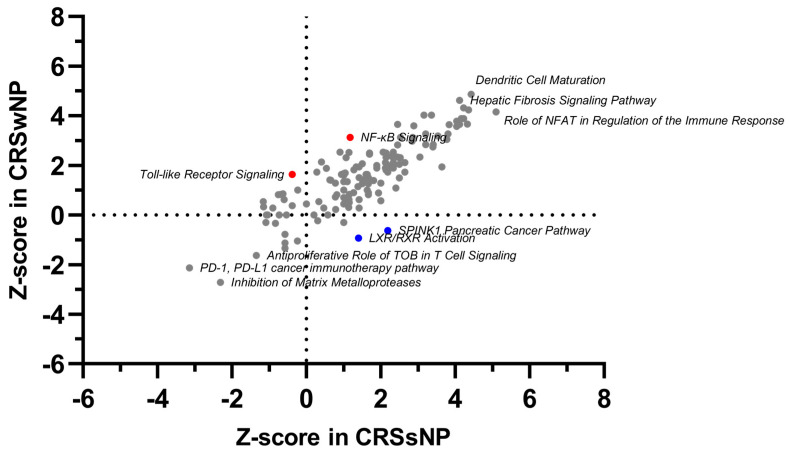
Comparison of enriched canonical pathways in CRSwNP and CRSsNP. The plot is based on the Z-score calculations obtained for a given set of genes using QIAGEN’s Ingenuity® Pathway Analysis (IPA®, QIAGEN, www.qiagen.com/ingenuity (accessed on 20 November 2021)). A positive Z-score represents strongly activated (Z-score > 2) pathways, and a negative Z-score represents significantly inhibited (Z-score < −2) pathways. Proposed unique phenotype-associated pathways are presented in red (●) for CRSwNP and in blue (●) for CRSsNP. Details in Appendix A (Canonical pathways).

**Figure 3 ijms-24-05541-f003:**
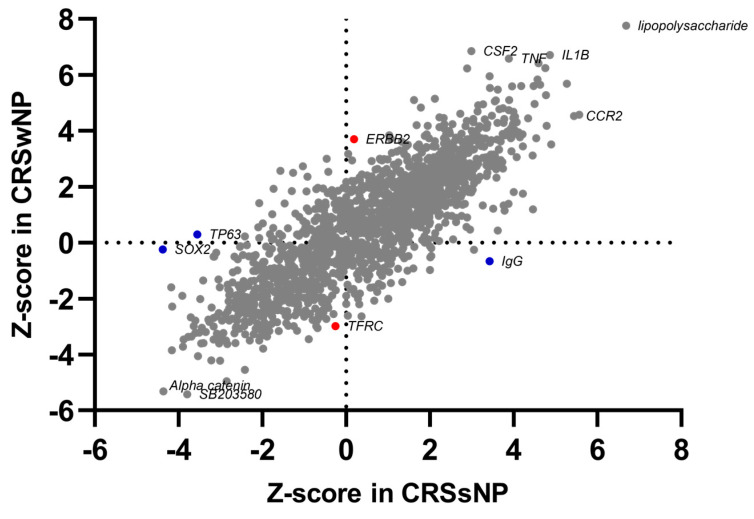
Comparison of enriched upstream regulators in CRSwNP and CRSsNP. The plot is based on the Z-score calculations obtained for a given set of genes, using the IPA analysis. A positive Z-score represents strongly activated (Z-score > 2) pathways, and a negative Z-score represents significantly inhibited (Z-score < −2) pathways. Proposed unique phenotype-associated pathways are presented in red (●) for CRSwNP and in blue (●) for CRSsNP. Details in Appendix A (Canonical pathways).

**Figure 4 ijms-24-05541-f004:**
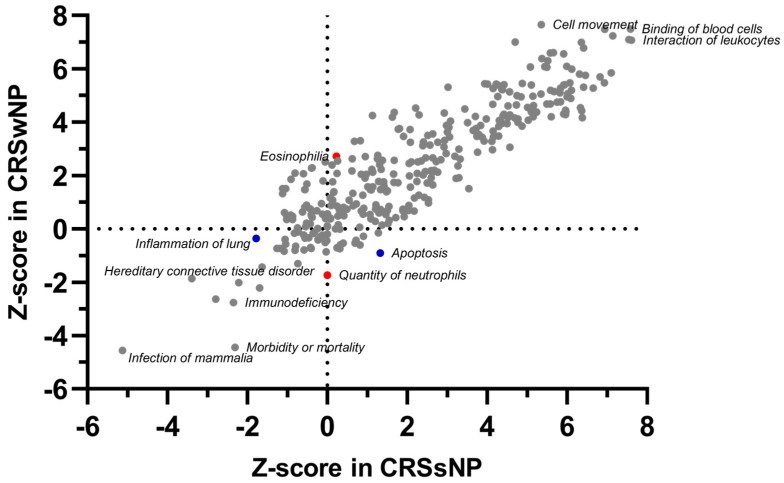
Comparison of enriched diseases and functions in CRSwNP and CRSsNP. The plot is based on the Z-score calculations obtained for a given set of genes, using the IPA analysis. A positive Z-score represents strongly activated (Z-score > 2) pathways, and a negative Z-score represents significantly inhibited (Z-score < −2) pathways. Proposed unique phenotype-associated pathways are presented in red (●) for CRSwNP and blue (●) for CRSsNP. Details in Appendix A (Diseases and functions).

**Figure 5 ijms-24-05541-f005:**
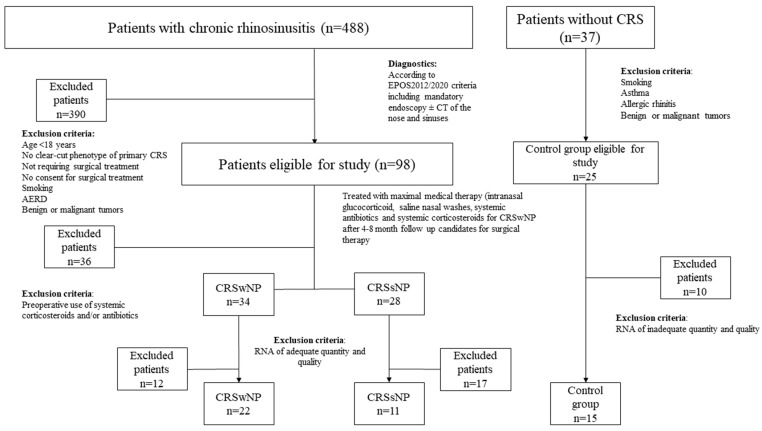
The inclusion flowchart.

**Figure 6 ijms-24-05541-f006:**
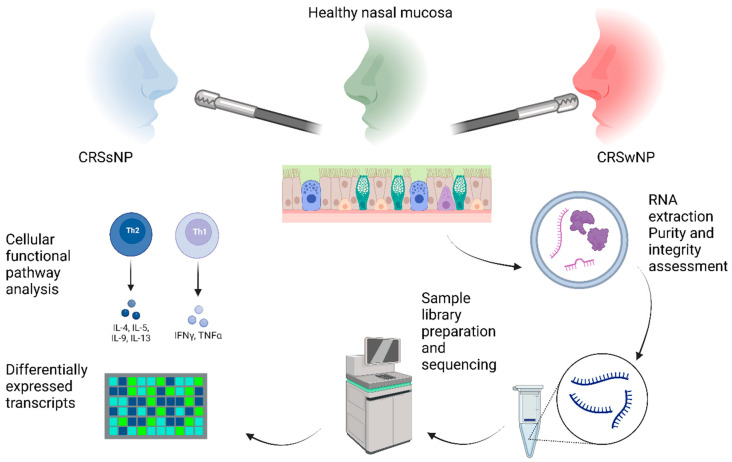
Graphical-study flowchart depicting several steps from initial tissue biopsy to RNA extraction, assessment of RNA quality, sample library preparation, and sequencing, with final stages of DEG and functional analysis using QIAGEN’s Ingenuity^®^ Pathway Analysis (IPA^®^, QIAGEN, www.qiagen.com/ingenuity (accessed on 20 November 2021)).

**Table 1 ijms-24-05541-t001:** Patients’ characteristics.

	Controls	Patients with CRSwNP	Patients with CRSsNP	*p*
Subjects, n (%)	15 (31)	22 (46)	11 (23)	-
Age (mean, min.–max., SD; years)	45.7 (18–79, SD 18.8)	54.2 (18–82, SD 16.9)	55.6 (33–78, SD 15.1)	0.21 ^1^
Females, n (%)	4 (27)	9 (41)	5 (46)	0.59 ^2^
Family history of CRS, n (%)	0 (0)	4 (18)	2 (18)	0.24 ^2^
Allergy, n (%)	-	12 (54)	4 (36)	0.46 ^3^
Asthma, n (%)	-	9 (41)	2 (18)	0.26 ^3^
GERD, n (%)	0 (0)	4 (18)	7 (64)	0.02 ^3^
GERD on therapy, n (%)	0 (0)	3 (14)	5 (45)	0.08 ^3^
Smell subjectively good, n (%)	15 (100)	11 (50)	11 (100)	<0.001 ^1^
Smell SS 12 score (mean, min.–max., SD)	9.56 (8–12, SD 1.5)	5.2 (2–7, SD 1.7)	9 (6–11, SD 1.8)	<0.001 ^1,5^
Patients with eosinophilia in tissue samples, n (%)	-	22 (100)	1 (9)	<0.001 ^3^
Eosinophilia in tissue samples (mean, min.–max., SD) No./HPF	-	30 (10–60, SD 12.2)	10	-
Patients with eosinophilia in blood, n (%)	-	8 (36)	1 (9)	0.21 ^3^
No. of past surgeries (mean, min.–max., SD)	-	0.6 (0–3, SD 0.9)	0.5 (0–2, SD 0.7)	0.72 ^4^
Time from last surgery (mean, min–max, SD; years)	-	5.8 (2–11, SD 3)	5.5 (0.8–16, SD 7)	0.33 ^4^
Patients with a need for systemic steroids (>2 times in past year), n (%)	-	10 (45)	-	-
Lund–Mackay score (range 0–24) mean, min.–max., SD)	-	18.1 (13–22, SD 2.2)	12.4 (7–20, SD 3.8)	<0.001 ^4^
Modified Kennedy–Lund score [21] (range 0–12), (mean, min.–max., SD)	-	9.7 (4–12, SD 2.4)	6 (2–8, SD 2.1)	<0.001 ^4^
Total polyp score (range 0–8) (mean, min.–max., SD)	-	5.9 (2–8, SD 2.2)	0	-
CRS symptoms VAS score (range 0–10); (mean, min.–max., SD)	4.9 (1–9, SD 2.3)	7.4 (4–9, SD 1.3)	6.3 (3–8, SD 1.4)	0.02 ^4^
SNOT-22 score (mean, min.–max., SD)	7.3 (2–15, SD 3.7)	43.9 (18–59, SD 12.3)	41.7 (21–70, SD 13.9)	0.31 ^4^
Patients with T2 clinical endotype, n (%)		22 (100)	1 (9)	<0.001 ^3^
Patients with non-T2 clinical endotype, n (%)		0	10 (91)	<0.001^3^

^1^ Kruskal–Wallis test, ^2^ Pearson Chi-Square Monte Carlo estimate (Confidence level 99%, Samples Number 10000), ^3^ Fisher’s exact test, ^4^ Mann–Whitney U test, ^5^ Control vs. Chronic rhinosinusitis without nasal polyps (CRSsNP), Mann–Whitney U test, *p* = 0.69, CRSwNP –chronic rhinosinusitis with nasal polyps, GERD—gastroesophageal reflux disease, Smell—subjective assessment of adequate smell sensation, SS 12– Sniffin’Sticks 12 test, Lund–Mackay Computer tomography (CT) score, Modified Kennedy–Lund endoscopic score, VAS score for subjective cumulative burden of all nasal symptoms, SNOT-22 validated Slovenian version of primary QoL questionnaire.

**Table 2 ijms-24-05541-t002:** Several unique phenotype genes are presented with their Log_2_ fold change values and false discovery rate (FDR), comparing CRSwNP/CRSsNP and controls. FDR values of 0.05 or less are outlined in bold. Genes outlined in red (●) are CRSwNP-unique, and those outlined in blue (●) are CRSsNP-unique. The rest are differentially expressed in both CRS phenotypes (CRSwNP and CRSsNP).

Full Gene Name	Gene	CRSsNP vs. Control—Log_2_ FC	CRSsNP vs. Control—FDR	CRSwNP vs. CRSsNP—Log_2_ FC	CRSwNP vs. CRSsNP—FDR	CRSwNP vs. Control—Log_2_ FC	CRSwNP vs. Control—FDR
Chloride Channel Accessory 1	* CLCA1 *	−1.85	3.56 × 10^−1^	10.52	**5.10 × 10^−12^**	8.70	**1.00 × 10^−10^**
Lipopolysaccharide-binding protein	*LBP*	9.32	**5.31 × 10^−13^**	−6.94	**4.11 × 10^−9^**	2.42	4.17 × 10^−2^
Lacritin	* LACRT *	−0.14	9.52 × 10^−1^	−6.53	**1.96 × 10^−8^**	−6.64	**1.00 × 10^−10^**
Pannexin 3	* PANX3 *	7.58	**9.51 × 10^−10^**	−5.43	**1.90 × 10^−9^**	2.10	2.13 × 10^−1^
C-X-C Motif Chemokine Ligand 13	* CXCL13 *	5.21	**2.19 × 10^−7^**	−5.33	**9.48 × 10^−10^**	−0.04	9.79 × 10^−1^
Lipocalin 1	* LCN1 *	−2.16	2.08 × 10^−1^	−5.08	**4.30 × 10^−7^**	−7.17	**1.35 × 10^−13^**
Prolactin-induced protein	* PIP *	−1.12	3.79 × 10^−1^	−4.87	**7.64 × 10^−7^**	−5.94	**1.00 × 10^−10^**
Cystatin D	* CST5 *	−1.99	1.84 × 10^−1^	−4.13	**7.76 × 10^−4^**	−6.11	**1.63 × 10^−11^**
Osteocrin	* OSTN *	4.96	**4.62 × 10^−5^**	−4.10	**9.03 × 10^−5^**	0.81	6.61 × 10^−1^
Matrix metallopeptidase 9	*MMP9*	5.07	**1.00 × 10^−10^**	−2.63	**3.92 × 10^−3^**	2.50	**2.92 × 10^−4^**
C-C motif chemokine ligand 7	*CCL7*	3.68	**1.22 × 10^−2^**	2.85	9.32 × 10^−2^	6.64	**3.58 × 10^−8^**
G protein-coupled receptor G7	* ADGRG7 *	0.38	8.65 × 10^−1^	6.00	**1.96 × 10^−4^**	6.67	**2.53 × 10^−7^**
Desmoglein−1	* DSG1 *	−9.24	**3.37 × 10^−8^**	6.24	**8.18 × 10^−4^**	−2.91	6.17 × 10^−2^
MT-RNR2 like 9 pseudogene	* MTRNR2L9 *	0.31	9.12 × 10^−1^	7.07	**1.40 × 10^−4^**	7.28	**5.26 × 10^−7^**
Small proline-rich protein 1A	* SPRR1A *	−8.33	**1.40 × 10^−13^**	7.89	**1.02 × 10^−10^**	−0.50	8.44 × 10^−1^
small proline-rich protein 2D	* SPRR2D *	−8.51	**1.27 × 10^−11^**	8.45	**1.45 × 10^−9^**	−0.06	9.84 × 10^−1^
Late cornified envelope 3D	* LCE3D *	−11.07	**1.18 × 10^−8^**	9.02	**1.87 × 10^−5^**	−1.96	3.48 × 10^−1^
Serine peptidase inhibitor kazal type 7	* SPINK7 *	−11.33	**1.00 × 10^−10^**	9.33	**8.05 × 10^−9^**	−1.95	3.77 × 10^−1^
Involucrin	* IVL *	−11.55	**1.00 × 10^−10^**	10.08	**5.94 × 10^−11^**	−1.48	5.03 × 10^−1^

**Table 3 ijms-24-05541-t003:** Cytokine-related genes and their expression presented with their Log_2_ fold change values and false discovery rate (FDR), comparing CRSwNP/CRSsNP and controls. FDR values of 0.05 or less are outlined in bold.

Endotypes ^1^	Cytokine ^1^	Full Gene Name	Gene	CRSsNP vs. Control—Log_2_ FC	CRSsNP vs. Control—FDR	CRSwNP vs. CRSsNP—Log_2_ FC	CRSwNP vs. CRSsNP—FDR	CRSwNP vs. Control—Log_2_ FC	CRSwNP vs. Control—FDR
T1	IFN-γ	Interferon gamma	*IFNG*	1.84	**2.12 × 10^−2^**	−1.49	1.10 × 10^−1^	0.39	7.52 × 10^−1^
	IL-12	Interleukin 12B	*IL12B*	2.16	**2.07 × 10^−2^**	−1.74	5.28 × 10^−2^	0.47	7.72 × 10^−1^
T2	IL-5	Interleukin 5	*IL5*	0.40	8.29 × 10^−1^	3.82	**2.57 × 10^−4^**	4.23	**4.66 × 10^−7^**
	IL-13	Interleukin 13	*IL13*	0.64	5.95 × 10^−1^	2.96	**1.81 × 10^−3^**	3.61	**1.49 × 10^−7^**
T3	IL-17	Interleukin 17B	*IL17B*	0.16	9.01 × 10^−1^	−2.24	**1.65 × 10^−2^**	−2.02	**3.27 × 10^−3^**

^1^ Endotypes and their cytokines as proposed by Staudacher et al. [2].

## Data Availability

Any additional supporting data beyond the contained and Appendix A are available from the corresponding author upon reasonable request. Data are contained within the article and Appendix A.

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
