# Peer review of "Transcriptomic Differentiation of Phenotypes in Chronic Rhinosinusitis and Its Implications for Understanding the Underlying Mechanisms"

_ijms, 2023, doi:10.3390/ijms24065541_

Round 1

Reviewer 1 Report

Authors acquired tissue samples from sinus of patients with CRSwNP and CRSsNP and extracted RNA from tissue samples. Then RNA data analysis was conducted to differentiate the specific genes between CRSsNP and CRSwNP.

Line 38-39: This sentence needs reference NO.  

Line 44-45: Identifying those mechanisms and their understanding -----illness: Reference is not necessary.

Line 47: Reference 9 showed no atopic dermatitis.

Line 48: this sentence “and reduce abundant possibilities” is not proper to understand the introduction. Or, please explain more to understand the abundant possibilitie.

Line 44-Line 61: Authors designed this study to assess transcriptomic analysis of genes in CRSwNP and CRSNP. Until now, many researchers have reported the expression of specific genes in CRSwNP and CRSsNP using various techniques.

Based on the previous results for the expression of genes specific to CRSwNP and CRSsNP, insufficient data evaluated by previous studies should be described in the introduction section, suggesting that the present study is necessary to fulfill complete data for specific genes in CRSwNP and CRSsNP.

Please describe the introduction more logically.

Line 75: Authors described that CRSwNP had s T2 endotype exclusively and CRSsNP was a predominantly non-T2 endotype.

      How do authors classify T2 endotype and non-T2 endotype?

      In this experiment, did authors conducted the analysis of cytokine expression in each phenotype? Because of cited No, it is confused to understand.

      Please explain in detail.

In Table2, abbreviated words for phenotype genes should be described as full names.

Figure 1 A; venn diagram;

         Do authors indicate that red circle means the genes of CRSwNP?

         Yellow circle indicates the gene of CRSsNP?

         Blue circle indicates those of controls?

         However, CRSwNP vs CRSsNP etc is confused to understand the meaning of venn diagram.

       1B: Which of phenotype does red dot indicate, CRSwNP or CERSwNP?

          Figure 1B is confused to understand what the red dot indicates.

For example, does red dot indicate specific genes for CRSwNP or CRSsNP?

Maybe, specific genes presented in Table2 are indicated by the red dot?

Line 162-163: Common key genes and pathways involved in CRS include---------.

       However, the data indicating “common key genes and pathways -----” were not found in the results section. Otherwise, do the authors cite these data form other reference?

Line 177-180: CRS-associated genes are----- lack sufficient evidence.

        These sentences are not the results of this paper. So, cited reference is required.

Line 187: Table E1 is not found in this manuscript.

Line 193: upregulated bone homeostasis genes, human bone sialoprotein gene IBSP (generally upregulated),

        Comma should be inserted between genes and human bone.

Line 200: supplementary table E3 is not found.

        Supplementary table S3 is found.

        Which is correct?

            The label of supplementary table should be corrected exactly.

Line 221: Woong Hwang et al should be changed to Hwang et al.

Line 273: Other CRSsNP downregulated genes were----

        : should be changed to “Other down regulated genes in CRSsNP -----“.

Line 298 ; no verb in this sentence.

Line 299: Ref 69 is not matched to a clear-cut phenotype.

Line 331. . [1] Is changed to [1].

Line 333-337 Sentences showed no verb.

Line 351;  ref 68 and 73 indicates that tissue sample sites is the ethmoid sinus cavities.

More exact sites are required in this study.

How do you differentiate the clustering into clinical non-T2 and T2 subtypes.

Do you use some techniques to differentiate non T2 and T2 subtypes?

Line 106; transcriptomic alterations in CRS-------

        Please describe in detail the evidence or data that authors come to these results.

        For examples, in line 112, as in CRSwNP immune regulation through NF-kB, Toll-like receptor------. How does authors come to this results?

        In fact, 2.3 (line 106-152) transcriptomic alterations in CRS-------. This paragraph is important results of this manuscript. Please describe the results again for readers to understand very well what authors means.

 Line 345: How many tissue samples did authors take from operation site?

         How many samples did authors use to extract the RNA?

         How many samples did authors use to analyze genes?

         Do authors repeat the gene analysis using the same samples?

Author Response

Dear Reviewer 1,

Thank you for the excellent remarks. The authors have made a thorough major revision of the proposed text in all elements brought forward by the reviewer. We have arranged comments and point-by-point responses below and incorporated them into the main text.

  1. Line 38-39: This sentence needs reference NO.

The authors agree with the reviewer and have accordingly inserted references.

  1. Line 44-45: Identifying those mechanisms and their understanding -----illness: Reference is not necessary.

The authors agree with the reviewer and have accordingly deleted the reference.

  1. Line 47: Reference 9 showed no atopic dermatitis.

The authors have referred to the paper published by Dong et al. Involvement and therapeutic implications of airway epithelial barrier dysfunction in type 2 inflammation of asthma, where authors associate Type 2 immunity with allergic rhinitis, allergic asthma, and atopic dermatitis.

The authors agree with the reviewer that additional references to Hammad and Lambrecht might be to solidify the association further.

  1. Line 48: this sentence “and reduce abundant possibilities” is not proper to understand the introduction. Or, please explain more to understand the abundant possibilitie.

The authors agree with the reviewer and have deleted the proposed part of the sentence. Since the problem usually becomes more evident with the data at hand, we agree it is premature to state it in the introduction. The abundant possibilities refer to all the possible molecular pathways exposed by genetic studies but lack exact evidence of having a degree of influence in CRS. To an extent, common diseases might help unravel some of the mechanisms.

  1. Line 44-Line 61: Authors designed this study to assess transcriptomic analysis of genes in CRSwNP and CRSNP. Until now, many researchers have reported the expression of specific genes in CRSwNP and CRSsNP using various techniques.

The authors agree with the reviewer that transcriptomic analysis is just one of the methods of analyzing molecular processes. Many authors have previously reported of clusters of genes or particular genes involved in CRS (both phenotypes). We have accordingly added three references.

  1. Based on the previous results for the expression of genes specific to CRSwNP and CRSsNP, insufficient data evaluated by previous studies should be described in the introduction section, suggesting that the present study is necessary to fulfill complete data for specific genes in CRSwNP and CRSsNP.

The authors agree that adequately assessing the need for additional research is essential. In other words, emphasize the need to confirm the previous results positively or negatively. We have added the appropriate text and reference to Orlandi et al. international consensus statement… where genes are pooled into subgroups, and most studies using various techniques were mentioned.

  1. Please describe the introduction more logically.

The authors have rewritten the introduction to make it more logically arranged.

  1. Line 75: Authors described that CRSwNP had s T2 endotype exclusively and CRSsNP was a predominantly non-T2 endotype.How do authors classify T2 endotype and non-T2 endotype?

In this experiment, did authors conducted the analysis of cytokine expression in each phenotype? Because of cited No, it is confused to understand.

Please explain in detail.

Thank you for the remark. Patients' clinical endotype were classified using the proposed criteria and cut-off points published in EPOS 2020 document using tissue eosinophilia, blood eosinophilia. Patients with values below the cut-off points were classified as non-T2 type. The authors agree that the analysis of cytokine expression would be incredibly welcome to support the clinical endotyping. But some other clinical data also allow valid associations between inflammatory endotypes and clinical presentations in CRS as shown by the results of Stevens et al.

The authors have additionally prepared Table 3 with the expression of cytokines in each phenotype. We have chosen cytokines for T1/T2/T3 endotype from a widely accepted paper of Staudacher et al. All data were extracted from Supplementary Table S1. Table 3 may confirm the validity of clinical T2/non-T2 assessment.

  1. In Table2, abbreviated words for phenotype genes should be described as full names.

The authors agree with the reviewer's proposal. Therefore we have added the full names to Table 2 accordingly.

  1. Figure 1 A; venn diagram; Do authors indicate that red circle means the genes of CRSwNP?

Thank you for the remark. The red circle annotates the differentially expressed genes of CRSwNP vs. CRSsNP.

  1. Yellow circle indicates the gene of CRSsNP?

The yellow circle indicates differentially expressed genes of CRSwNP vs. Controls.

  1. Blue circle indicates those of controls?

The blue circle represents the differentially expressed genes of CRSsNP vs. Controls.

  1. However, CRSwNP vs CRSsNP etc is confused to understand the meaning of venn diagram.

The red circle, CRSwNP vs. CRSsNP, shows differentially expressed genes comparing CRSwNP and CRSsNP.

  1. 1B: Which of phenotype does red dot indicate, CRSwNP or CERSwNP?

Thank you for the remark. The red dots in Figure 1B indicates genes with the highest differential expression. DEGs marked with red dots with Log2 FC near 0 on the x axis,

represents DEGs with the highest differential expression in CRSwNP with red dots and DEGS with the highest differential expression in CSRsNP with blue dots.

We agree with the reviewer that this can be somehow confusing. Therefore in the revised version, we marked the DEGs with the highest differential expression in CRSwNP with red dots and DEGS with the highest differential expression in CSRsNP with blue dots.

  1. Figure 1B is confused to understand what the red dot indicates.

We agree with the reviewer’s comment that the red dots were somehow confusing. Therefore we have corrected Table 1B and have marked DEGs with the highest differential expression in CRSwNP with red dots and DEGS with the highest differential expression in CSRsNP with blue dots.

Besides, we have also added a sentence in the legend further explaining this, stating, “The proposed unique phenotype-associated DEGs with highest differential expression in CRSwNP are presented in red (●) and CRSsNP in blue ().”

Furthermore, to adhere to the same style and to avoid any possible confusion, we have similarly changed the red dots into red and blue dots in Figures 2, 3 and 4.

  1. For example, does red dot indicate specific genes for CRSwNP or CRSsNP?

As mentioned above, we agree with the reviewer and have corrected Figure 2B

  1. Maybe, specific genes presented in Table2 are indicated by the red dot?

Thank you for pointing this out. Selected unique phenotype genes presented in Figure 1B are also pointied out in Table 2. We agree with the reviewer’s suggestion and to avoid anyl possible confusion we have (similarly as in Figure 1B) marked CRSwNP specific DEGs in red and CRSsNP specific DEGs in blue..

  1. Line 162-163: Common key genes and pathways involved in CRS include---------. However, the data indicating “common key genes and pathways -----” were not found in the results section. Otherwise, do the authors cite these data form other reference?

Thank you for the remark. The authors agree with the reviewer that the presentation may lack a clearer perspective, therefore, we propose a change in the Discussion section and add the reference to Figure 2 and Supplementary Table S2). In the Discussion section we have stated common genes and pathways involved in CRS, including dendritic cell maturation, neuroinflammation pathway, and inhibition of matrix metalloproteinases. Those are the canonical pathways marked with a grey dot in Figure 2.

  1. Line 177-180: CRS-associated genes are----- lack sufficient evidence. These sentences are not the results of this paper. So, cited reference is required.

The authors agree with the reviewer. The part of the discussion where we state the pooling of the CRS- associated genes is not a result of the proposed paper. We have therefore cited both excellent meta-analyses from EPOS 2020 and ICAR-RS-2021.

During the comprehensive research, we identified numerous studies using different techniques, proposing CRS-related genes and explaining their possible role – as previously remarked by the reviewer and agreed upon by the authors. Authors think both meta-analyses made by Fokkens et al. and Orlandi et al. are excellent and can be easily accessed by the readers. We also agree with the reviewer about the need for an additional reference for the strength of evidence about gene pooling and have added the ICAR-RS-2021 reference to the first sentence.  

  1. Line 187: Table E1 is not found in this manuscript.

The authors agree with the reviewer. Indeed Supplementary Table E3 is not part of the paper. It was an unfortunate typing error. Therefore we have changed Supplementary Table E3 to Supplementary Table S3.

  1. Line 193: upregulated bone homeostasis genes, human bone sialoprotein gene IBSP (generally upregulated),Comma should be inserted between genes and human bone.

The authors agree with the reviewer. Therefore proposed changes have been made in the text.

  1. Line 200: supplementary table E3 is not found.

The authors agree with the reviewer. Indeed Supplementary Table E3 is not part of the paper. Therefore we have changed Supplementary Table E3 to Supplementary Table S3.

  1. Supplementary table S3 is found. Which is correct? The label of supplementary table should be corrected exactly.

Thank you for the remark. The authors agree Supplementary Table S3 is correct and have made changes throughout the proposed paper for all the Supplementary Tables.

  1. Line 221: Woong Hwang et al should be changed to Hwang et al.

The authors agree with the reviewer and have changed the referred author accordingly.

  1. Line 273: Other CRSsNP downregulated genes were----        : should be changed to “Other down regulated genes in CRSsNP -----“.

The authors agree with the reviewer and have implemented changes accordingly.

  1. Line 298 ; no verb in this sentence.

The authors agree with the reviewer and have implemented changes accordingly.

  1. Line 299: Ref 69 is not matched to a clear-cut phenotype.

Thank you for the remark. The authors have changed the references accordingly. One to general document as EPOS 2020, the other one on the paper of Staudacher et al. Use of endotypes, phenotypes…

  1. Line 331. . [1] Is changed to [1].

The authors have changed the text accordingly.

  1. Line 333-337 Sentences showed no verb.

The authors have changed the text accordingly.

  1. Line 351;  ref 68 and 73 indicates that tissue sample sites is the ethmoid sinus cavities. More exact sites are required in this study.

The authors used tissue from the sinonasal mucosa of the middle meatus or adjacent sinonasal region. We agree with the reviewer that previous studies (referred to in the proposed paper) have addressed the problem of the exact site within the narrow space of the middle meatus. Our main goal when performing the biopsy in patients with CRSwNP was to get the mucosa from the distal part of the polyp mass, therefore, from the attachment of the polyp in the anterior ethmoid. In CRSsNP, the diseased mucosa of the middle meatus, the anterior ethmoid or bulla ethmoidalis.

  1. How do you differentiate the clustering into clinical non-T2 and T2 subtypes.Do you use some techniques to differentiate non T2 and T2 subtypes?

Thank you for the remark. In the past, it has been difficult to attain any significant clinical clustering using the routinely available data in a clinical setting. As you stated in previous remarks, the cytokine expression analysis might ideally define the T2 substrate in CRSwNP and CRSsNP phenotypes. But other researchers and consensus documents, as they confirm the stated, also propose clinical guidelines based on available data as Stevens et al. Associations between inflammatory endotypes and clinical presentations in CRS. We use tissue eosinophilia, blood eosinophilia, and the presence of asthma. Our dominant method is tissue eosinophilia.

  1. Line 106; transcriptomic alterations in CRS-------Please describe in detail the evidence or data that authors come to these results.For examples, in line 112, as in CRSwNP immune regulation through NF-kB, Toll-like receptor------. How does authors come to this results?

The authors have performed a network and pathway analysis of the top differentially expressed genes (DEG) using the Quiagen’s Ingenuity Pathway Analysis (IPA) unraveling the mechanisms behind the genes found in the DEGs (Venn diagram, Figure 1A,1B). For instance, NF-kB signaling had Z-value of 3.13 in tissue of CRSwNP vs. a Z-value of 1.18 in CRSwNP. Toll-like receptor signaling had a positive Z-value in CRSwNP and negative in CRSsNP.  In fact, it is a further analyzed part of the data gathered.

  1. In fact, 2.3 (line 106-152) transcriptomic alterations in CRS-------. This paragraph is important results of this manuscript. Please describe the results again for readers to understand very well what authors means.

The authors concur with the reviewer and have added an additional description of the results within the paragraph.

  1. Line 345: How many tissue samples did authors take from operation site?

62 CRS and 25 control samples were taken from the operation/biopsy site for intended RNA extraction. Another 62 samples from the same site were taken for pathohistological examination.

  1. How many samples did authors use to extract the RNA?

48 samples were taken to extract RNA.

  1. How many samples did authors use to analyze genes?

The authors have taken two separate samples of tissue (one for RNA and the other for pathohistolgical examination) from the same site in the same patient. Altogether 48 samples were taken for RNA extraction. Out of them, as evident Figure 1, in 48 samples (22 CRSwNP, 11 CRSsNP, 15 Controls) were taken for RNAseq analysis.

  1. Do authors repeat the gene analysis using the same samples?

No, we did not repeat the gene analysis using the same samples

Reviewer 2 Report

This well-written study by a research group from Slovenia presents the results of an analysis of the nasal tissue transcriptome in patients suffering from chronic rhinosinusitis (CRS). The authors carried out RNA-sequencing of subjects with either CRS with polyps, without polyps, or healthy controls. Methodological assessments included differently expressed genes (DEGs) as well as functional and pathway analysis. Hundreds of DEGs were identified for CRS in general, with approximately half being specific for CRSwNP and CRSsNP, respectively. Specific involvement of dendritic cell maturation, neuroinflammation, and inhibition of matrix metalloproteinases was highlighted. In CRSwNP, NF-kβ, TLR, HIF1α, and Th2 appear to play major roles, while in CRSsNP, primarily NFAT and the Calcium pathway seemed to be relevant. The authors suggest that these findings lead to a better understanding of the pathophysiology of CRS and might lead to novel treatment strategies. While overall already in good condition, with sound methodology and good presentation of results, the major questions that remain are whether the limited novel insights that are added to the literature here are worth publishing (as multiple groups have already looked at RNA-seq in CRS), and why the methodology describes the exclusion of so many patients due to inadequate quality and quantity of RNA? In addition, minor edits can improve the manuscript. 

Abstract: 

- The authors already define CRS in line 15 and then still use the term “chronic rhinosinusitis” in line 18. Once you define an abbreviation, please use it throughout the manuscript.

- CRSsNP and CRSwNP are commonly used in the CRS literature, but have not been defined in the abstract.

Introduction:

Line 49: “…understanding of the CRS…” should rather be “…understanding of CRS…”

Results:
Table 1: There is a typo regarding smell in the CRSwNP column: “SD 1,7” should be “SD 1.7”

Table 1 legend: Shouldn’t it be “Lund Mackay Score” instead of “Lund MacKay Score”?

Materials and Methods:

Line 333-337: This sentence is incomplete: “As well as computer…Slovenian SNOT-22 score.”

Figure 5: My concern is that the exclusion of patients due to inadequate quality and quantity of RNA is incredibly high. E.g., in the CRSsNP group, 17 patients had to be removed out of 28 samples individuals, resulting in only 11 valid data points. As this is fresh tissue that is being processed, I would expect a much higher success rate.

Author Response

Dear Reviewer 2,

Thank you for the excellent remarks and thorough review. We have arranged comments and point-by-point responses below and incorporated them into the main text.

This well-written study by a research group from Slovenia presents the results of an analysis of the nasal tissue transcriptome in patients suffering from chronic rhinosinusitis (CRS). The authors carried out RNA-sequencing of subjects with either CRS with polyps, without polyps, or healthy controls. Methodological assessments included differently expressed genes (DEGs) as well as functional and pathway analysis. Hundreds of DEGs were identified for CRS in general, with approximately half being specific for CRSwNP and CRSsNP, respectively. Specific involvement of dendritic cell maturation, neuroinflammation, and inhibition of matrix metalloproteinases was highlighted. In CRSwNP, NF-kβ, TLR, HIF1α, and Th2 appear to play major roles, while in CRSsNP, primarily NFAT and the Calcium pathway seemed to be relevant. The authors suggest that these findings lead to a better understanding of the pathophysiology of CRS and might lead to novel treatment strategies.

  1. While overall already in good condition, with sound methodology and good presentation of results, the major questions that remain are whether the limited novel insights that are added to the literature here are worth publishing (as multiple groups have already looked at RNA-seq in CRS), and why the methodology describes the exclusion of so many patients due to inadequate quality and quantity of RNA? In addition, minor edits can improve the manuscript.

The authors thank the reviewer for an honest and straightforward review. As we agree that the methodology using RNA-seq has been used in CRS, the results of the genetic basics in understanding the CRS, especially both phenotypes, are not precise. Furthermore, the studies performed were based in opposite geographical locations, the latter being extremely important in already known overlapping of some pathophysiological features as cytokine expression within and outside the same geographical area. Regarding both major CRS consensus documents EPOS 2020 and ICAR-RS-2021 from Europe and the United States, respectively, regarding genes involved in CRS, there still are unmet needs in finding and confirming the CRS genes. The authors share that further research may go towards more comprehensive recruitment of patients in combined multicentric studies or meta-analysis of present and new studies, especially combining different molecular methods aiming for the same vital genes, pathways, and regulators.

Abstract: 

  1. - The authors already define CRS in line 15 and then still use the term “chronic rhinosinusitis” in line 18. Once you define an abbreviation, please use it throughout the manuscript.

The authors agree and thank the reviewer for the remark. The authors have made the changes in the text.

  1. - CRSsNP and CRSwNP are commonly used in the CRS literature, but have not been defined in the abstract.

The authors agree and thank the reviewer for the remark. The authors have made the changes in the text.

Introduction:

  1. Line 49: “…understanding of the CRS…” should rather be “…understanding of CRS…”

The authors agree and thank the reviewer for the remark. The authors have made the changes in the text.

Results:

  1. Table 1: There is a typo regarding smell in the CRSwNP column: “SD 1,7” should be “SD 1.7”

The authors agree and thank the reviewer for the remark. The authors have made the changes in the text.

  1. Table 1 legend: Shouldn’t it be “Lund Mackay Score” instead of “Lund MacKay Score”?

The authors agree and thank the reviewer for the remark. The authors have made the changes in the text.

Materials and Methods:

  1. Line 333-337: This sentence is incomplete: “As well as computer…Slovenian SNOT-22 score.”

The authors agree and thank the reviewer for the remark. The authors have made the changes in the text.

  1. Figure 5: My concern is that the exclusion of patients due to inadequate quality and quantity of RNA is incredibly high. E.g., in the CRSsNP group, 17 patients had to be removed out of 28 samples individuals, resulting in only 11 valid data points. As this is fresh tissue that is being processed, I would expect a much higher success rate.

The authors agree with the reviewer about the fail rate of tissue specimens with adequate RNA quality and quantity. We addressed the concern further as one of the valid study limitations.

Round 2

Reviewer 1 Report

I think that authors have well corrected the issues reviewer indicated. 

Reviewer 2 Report

The authors have adequately addressed all my comments.